# Non-Nucleoside Reverse Transcriptase Inhibitors Join Forces with Integrase Inhibitors to Combat HIV

**DOI:** 10.3390/ph13060122

**Published:** 2020-06-11

**Authors:** Daniel M. Himmel, Eddy Arnold

**Affiliations:** 1Himmel Sci Med Com, L.L.C., Bala Cynwyd, PA 19004, USA; 2Center for Advanced Biotechnology and Medicine (CABM), Department of Chemistry and Chemical Biology, Rutgers University, Piscataway, NJ 08854, USA; arnold@cabm.rutgers.edu

**Keywords:** non-nucleoside reverse transcriptase inhibitor, etravirine, rilpivirine, integrase strand transfer inhibitor, dolutegravir, adherence, drug resistance, dual therapy, long-acting therapy, HIV

## Abstract

In the treatment of acquired immune deficiency syndrome (AIDS), the diarylpyrimidine (DAPY) analogs etravirine (ETR) and rilpivirine (RPV) have been widely effective against human immunodeficiency virus (HIV) variants that are resistant to other non-nucleoside reverse transcriptase inhibitors (NNRTIs). With non-inferior or improved efficacy, better safety profiles, and lower doses or pill burdens than other NNRTIs in the clinic, combination therapies including either of these two drugs have led to higher adherence than other NNRTI-containing treatments. In a separate development, HIV integrase strand transfer inhibitors (INSTIs) have shown efficacy in treating AIDS, including raltegravir (RAL), elvitegravir (EVG), cabotegravir (CAB), bictegravir (BIC), and dolutegravir (DTG). Of these, DTG and BIC perform better against a wide range of resistance mutations than other INSTIs. Nevertheless, drug-resistant combinations of mutations have begun to emerge against all DAPYs and INSTIs, attributable in part to non-adherence. New dual therapies that may promote better adherence combine ETR or RPV with an INSTI and have been safer and non-inferior to more traditional triple-drug treatments. Long-acting dual- and triple-therapies combining ETR or RPV with INSTIs are under study and may further improve adherence. Here, highly resistant emergent mutations and efficacy data on these novel treatments are reviewed. Overall, ETR or RPV, in combination with INSTIs, may be treatments of choice as long-term maintenance therapies that optimize efficacy, adherence, and safety.

## 1. Introduction

In spite of considerable success in the treatment of acquired immune deficiency syndrome (AIDS), combination antiretroviral therapies (cARTs), also known as highly active antiretroviral therapies (HAARTs), still face challenges in long-term safety, regimen adherence, and the emergence of drug resistance. In the cART approach, a cocktail of anti-AIDS drugs target functions that are essential to the life cycle of human immunodeficiency virus (HIV) [1], which causes AIDS [2,3,4,5,6,7,8,9,10]. Anti-AIDS treatment is necessarily lifelong, because HIV integrates its genomic material (in DNA form) into the chromosomes of CD4^+^ T-cells soon after infecting a patient, which creates a pool of latently infected cells that cannot be eliminated by currently available treatments [11,12]. Historically, a key goal of treatment has been to maintain a low viral load in the plasma. More recently, the “Undetectable Equals Untransmittable” (U = U) objective has been widely adopted, based on evidence that HIV is not transmittable from patients whose viremia has dropped to undetectable levels [13,14]. To achieve that objective, most cART regimens target at least two of three HIV enzymes that are essential for viral replication: reverse transcriptase (RT), integrase (IN), and protease (PR) [15,16,17]. RT converts the single-stranded viral RNA (ssRNA) to double-stranded DNA (dsDNA) [18], and IN facilitates the integration of this dsDNA into the host genome [19,20,21]. The dsDNA codes for two polyproteins (Gag and Gag-Pol) that contain the immature structural proteins (Gag) and the enzymes (Pol), which HIV PR processes into their mature counterparts, including RT, IN, and PR [22,23,24,25,26,27]. For patients with multi-class, drug-resistant HIV infections, other stages of the HIV life cycle have been targeted, such as the CCR5 receptor and gp41-mediated viral fusion to the host cell [11,15]. Historically, first-line cARTs have generally consisted of two nucleoside/nucleotide RT inhibitors (NRTIs) and one inhibitor in another class, such as a non-nucleoside RT inhibitor (NNRTI), an IN strand transfer inhibitor (INSTI), or a pharmacologically-boosted protease inhibitor (PI) [15,28,29]. New formulations are needed that are safer and support greater adherence, because long-term treatment with cART has been associated with a variety of comorbidities [30,31,32,33,34,35,36,37,38,39], as well as non-adherence that leads to drug resistance and virologic failure [40,41]. Etravirine (ETR) and rilpivirine (RPV) are two recently introduced NNRTI therapeutic agents (Figure 1a) with improved adherence that have been widely effective (Table 1) against HIV variants resistant to previous-generation NNRTIs, such as efavirenz (EFV), nevirapine (NVP), and delavirdine (DLV) [15,16,29,42,43,44]. New therapies have been investigated or approved that combine ETR or RPV with an INSTI, such as raltegravir (RAL), cabotegravir (CAB), or dolutegravir (DTG, Figure 1b). This article reviews the efficacy, safety, adherence, a summary of emergent drug resistance, and possible future trends for these NNRTI/INSTI cARTs in the treatment of AIDS.

## 2. Efficacy of ETR and RPV

ETR and RPV (Figure 1a) are part of a family of diarylpyrimidine (DAPY) compounds that were developed in the 1990s and early 2000s as part of a multidisciplinary, structure-based drug design collaboration between Janssen Pharmaceutica, Tibotec, and the Arnold laboratory [29,44]. ETR was approved in 2008 [45] and RPV, in 2011 [46]. Clinically, both compounds had good safety profiles [47,48], were safer and non-inferior to EFV [49,50,51,52,53,54], and were effective at suppressing viremia bearing common NNRTI-resistance mutations (Table 1) [16,55,56,57,58,59]. In the SENSE Phase II trial, the efficacy of ETR and EFV were comparable, and the adverse effects (AEs) frequency that was seen with ETR patients was similar or lower than that of EFV subjects, including significantly fewer Grade 2–4 neuropsychiatric AEs (*p* < 0.01) [53]. The ECHO/THRIVE Phase III trials found that RPV had a better safety profile than EFV, being characterized by a significantly lower incidence of treatment-related Grade 2–4 AEs (*p* < 0.0001) [47,51,52]. As a cART component, RPV showed efficacy that was comparable to EFV [47,49,50,52,60] or better [51], based on response (number of patients achieving a viral load <50 plasma HIV-1 RNA copies/mL).

A somewhat higher virologic failure rate was seen with RPV treatment than with EFV, but statistical significance was not established [47,50,52,60]. However, among RPV patients with a high baseline viral load (>100,000 HIV-1 RNA copies/mL) or a low baseline CD4^+^ T-cell count (<200 cells/µL), the virologic failure rate was significantly higher when compared to EFV patients [49,54,61], and the United States Food and Drug Administration (FDA) has indicated RPV for use only by patients with viremia ≤100,000 HIV-1 RNA copies/mL at the start of treatment [50,62,63]. In the SENSE Phase II trial, the virologic failure rate with ETR was comparable or lower than with EFV [53]. The virologic failure of patients on ETR, RPV, and cARTs has been attributed in part to regimen non-adherence [40,41,50,60,64,65].

## 3. DAPY/INSTI Dual Therapies

Dual therapies have begun to replace other cARTs as a maintenance strategy for improving adherence and safety. A number of dual therapies that combine ETR or RPV with an INSTI are currently available or under investigation. These include ETR/RAL [73,74], RPV/CAB [11,75], and RPV/DTG [62]. Good efficacy and safety profiles have characterized these therapies.

### 3.1. ETR/RAL

Four observational studies have reported efficacy for ETR/RAL therapies administered to HIV patients who were previously on other cARTs, but the criteria for efficacy differed with each study. A retrospective study in Paris, France with 18 patients defined efficacy as a plasma HIV load <200 copies/mL and saw 83.3% efficacy after 12 months of the ETR/RAL treatment [76]. In a prospective study conducted in Barcelona, Spain, the criteria for efficacy was a viral load <50 copies/mL, achieved by 84% of 25 patients at 48 weeks [74]. This finding was corroborated by a prospective observational study in Madrid, Spain, in which 96% of 25 patients were still virologically suppressed 48 weeks after switching to ETR/RAL dual therapy [77]. At the University of Bologna, Italy, a prospective study with 38 patients used a tighter viral load criteria of <20 copies/mL, and 81.6% of patients achieved this outcome at 48 weeks [73]. The treatment was well tolerated with low incidence of serious AEs in these studies. While these outcomes suggest that ETR/RAL is a viable treatment option, a larger controlled study is needed in order to validate the findings.

### 3.2. RPV/CAB

The Phase IIb LATTE trial in Canada and the United States compared the efficacy and safety of RPV/CAB to that of a traditional triple-drug therapy cART consisting of EFV and two NRTIs. The study began with a head-to-head comparison between CAB and EFV for 24 weeks. During this “induction therapy” period, the 243 patients, all on two NRTIs, were randomized between 600 mg EFV and CAB at doses of 10 mg, 30 mg, or 60 mg. By week 24, approximately 86% of each of the CAB groups and 74% of EFV patients had achieved the virologic suppression endpoint of <50 viral RNA copies/mL. At week 24 onward, virologically suppressed patients in the CAB groups stopped taking the NRTIs and received 25 mg RPV in addition to their randomly assigned CAB dose, while virologically suppressed EFV patients continued with their triple therapy regimen. At week 48, 82% of the RPV/CAB patients and 71% of the EFV/NRTs patients were virologically suppressed. At week 96, 76% of the RPV/CAB patients and 63% of the EFV/NRTs patients were still virologically suppressed. The AEs were lower in the RPV/CAB groups (51%) than the EFV/NRTs group (68%). There were six (3%) AE-related withdrawals from the study in the RPV/CAB groups and nine (15%) in the EFV/NRTs group [75]. The results suggested that the RPV/CAB dual therapy was at least as effective as the EFV/NRTs triple therapy and possibly safer. This was not a blinded study.

### 3.3. RPV/DTG

RPV/DTG was the first dual therapy regimen to be approved by the FDA for the maintenance treatment of virologically suppressed patients with HIV-1 [62]. The efficacy and safety of RPV/DTG (for review, see ref. [62]) were investigated in the SWORD-1 and -2 trials [78]. These Phase III, open-label, multicenter, randomized parallel-group trials compared pooled data for 512 adults on various conventional cARTs (active control) and 516 adults who switched from other therapies to a once/daily pill of DTG (50 mg) and RPV (25 mg). At baseline, the two groups had stable suppressed viremia (<50 copies per mL) and they were demographically well matched, including the proportion of patients taking various therapeutic agents (INSTIs, NNRTIs, PIs) and CD4 counts. At week 48 in the intention-to-treat population, 95% of both groups had achieved viral loads under 50 copies/mL with an adjusted treatment difference of −0.2% (95% CI −3.0 to 2.5). AE incidences were 77% in the RPV/DTG group and 71% in the conventional cARTs group. AE-related discontinuations comprised 3% of RPV/DTG patients and <1% of the conventional cARTs patients. Although these trials suggested non-inferiority of the RPV/DTG treatment, bias related to the open-label design of the studies could not be ruled out.

These results were supported by three observational studies in which virologic suppression (<50 copies/mL) was maintained in patients who switched from conventional cARTs to RPV/DTG therapy. A prospective study observed the same proportion of virologic suppression (91.4%) at week 48 and baseline for 35 patients who had switched from other cARTs to RPV/DTG, with no difference in safety profile [79]. In a study following 145 cART-experienced patients who switched to RPV/DTG treatment [80], 138 patients (95.2%) had <50 copies/mL at week 96 versus 123 patients (85%) at baseline. In a retrospective analysis of 152 virologically suppressed patients who switched from various cARTs to RPV/DTG [81], virologic suppression continued in 99.1% of the patients at week 24, with three virologic failures (2%), twelve (8%) AE-related discontinuations, no Grade 3–4 AEs, and no significant change in CD4 or CD8 counts.

A number of analyses have compared DTG to RAL, EFV/tenofovir disoproxil fumarate/emtricitabine, or the PI combination of darunavir/ritonavir. A 48-week, Phase III, randomized, double-blinded, active-controlled study (SAILING) with 715 patients in virologic failure (≥400 plasma HIV-1 RNA copies/mL) favored DTG over RAL in the percent of patients achieving an endpoint of <50 HIV-1 RNA copies/mL (adjusted difference 7.4%, 95% confidence interval: 0.7–14.2; *p* = 0.03), with similar safety outcomes [82]. A 96-week Phase III, randomized, double-blinded, active-controlled study (SPRING-2) with 822 patients found DTG to be non-inferior to RAL in percent efficacy (adjusted difference 4.5%, 95% confidence interval: −1.1–10.0%), with fewer virologic failures in the DTG group (5%) than the RAL group (10%), and similar safety outcomes [83]. These results were further validated by a larger analysis that combined 48-week data from the SPRING-2 study (*n* = 822), the SINGLE study (DTG/abacavir/lamivudine versus EFV/tenofovir disoproxil fumarate/emtricitabine, *n* = 833), and the FLAMINGO study (DTG versus darunavir/ritonavir, *n* = 484). While SPRING-2 and SINGLE were double-blinded studies, FLAMINGO was an open-label study. In all three studies, more DTG patients achieved the primary endpoint (<50 plasma HIV-1 RNA copies/mL) than the patients on the active control, a result that was significant in both the SINGLE and FLAMINGO trials [84,85,86,87]. The studies support the view that DTG is non-inferior or possibly superior to RAL and superior to EFV/tenofovir disoproxil fumarate/emtricitabine in efficacy.

No head-to-head comparisons between RPV/DTG and other NNRTI/INSTI dual therapies have been published as of 2019. One real-life, observational case report followed seven patients who maintained virologic suppression (HIV RNA <50 copies/mL) for 24 weeks after successfully switching from ETR/RAL to RPV/DTG to reduce pill burden and avoid the toxicity of previous treatments [88]. Motivations for changing from other treatments to RPV/DTG therapy have included treatment simplification, drug interactions, persistent low-level viremia, non-adherence, virologic failure, and toxicity [62,80,81].

## 4. Safety

Long-term cARTs face a variety of safety challenges from associated comorbidities [30,33,39,89]. Patients on cARTs, including those with NRTIs, NNRTIs, and/or PIs, tend to suffer bone mineral density loss, including osteoporosis and increased risk of bone fractures. The site and extent of bone mineral density loss depends on the choice of therapeutic agents [35]. Nephrotoxicity is often seen with NRTIs and certain PIs, such as atazanavir (ATV) [31,32,36], while some INSTIs, including RAL and DTG, have been associated with a non-progressive, non-pathological reduction in tubular uptake of serum creatinine [90]. Some NRTIs and PIs have been associated with long-term cardiovascular toxicity [34,37,38,89], lipodystrophy [30], or mitochondrial toxicity that results in peripheral neuropathy, myopathy, hepatic failure, or lactic acidosis [30,39,91]. ATV/RAL and ATV/tenofovir disoproxil fumarate/emtricitabine therapies have been associated with hyperbilirubinemia [33]. Because anti-AIDS treatment is lifelong, treatment formulations are needed that support improved safety and adherence.

### 4.1. Neuropsychiatric Comorbidities

A recent review [89] stated that there are conflicting data on whether EFV treatment is associated with a higher incidence of suicidal tendency than other treatments. On closer scrutiny, the data are not actually in conflict. An analysis of four studies in which 5332 patients were randomized between EFV-containing and EFV-free cARTs concluded that suicidal tendency was significantly more likely in the EFV groups, with a hazard ratio of 2.28 (95% confidence interval: 1.27–4.10, *p* = 0.006) for suicidal ideation, attempt, or completion. However, three of the four studies were open-label, and the studies did not share a standardized questionnaire on suicidal tendency [92]. A “Multi-Item Gamma Poisson Shrinker” disproportionality analysis of the FDA Adverse Event Reporting System reported “no evident association between efavirenz use and suicidality.” The analysis compared EFV to ETR, RAL, nevirapine, ATV, and darunavir. As a positive control, two unrelated antidepressants that were known to be associated with suicidality were included in the analysis. This study used a pre-determined threshold of twice as many suicide-related events as other drug-adverse-event combinations. To be sure, the two positive controls exceeded the pre-determined threshold for suicidal tendency, whereas none of the six anti-AIDS drugs did. However, in three of the four measures for suicide tendency (suicide ideation, attempt, completion, combined data), events among EFV patients were more numerous than all other anti-AIDS drugs in the comparison. Statistical significance was not reported. The authors cautioned that their study should not be misinterpreted as an absence of association between EFV and suicidal tendency [93]. These studies suggest that suicidal tendency might be higher with EFV than many other anti-AIDS agents, but a carefully controlled blinded study has yet to be conducted in order to clarify the significance and unbiased hazard ratios.

Notwithstanding questions about the magnitude of the suicidal risk that is associated with EFV, a reduction in frequency of neuropsychiatric comorbidities is likely upon switching to RPV or ETR. Substituting RPV for EFV significantly lowered the incidence of neurological AEs for treatment-naïve HIV-1 patients in the ECHO/THRIVE trials (*p* < 0.0001) [47,94]. In agreement with this finding, the Agence Nationale de Recherches sur le SIDA et les Hépatites Virales (ANRS) CO3 Aquitaine Cohort study [95] reported that 47.7% of 86 patients who switched from an EFV-containing treatment to an RPV/emtricitabine/tenofovir therapy experienced a decrease in neuropsychiatric AEs. Likewise, a Phase II, double-blinded study randomizing 157 patients between ETR-containing and EFV-containing cARTs found significantly fewer neuropsychiatric AEs (*p* = 0.011) in the ETR group (6.3%) than the EFV group (21.5%) [53].

Questions have also been raised regarding DTG association with neurological comorbidities. In a Netherlands-based cohort study, a 13.7% DTG intolerance rate was observed among 556 patients, especially if they also were administered abacavir. Comorbidities in these patients included neuropsychiatric (anxiety, depression, insomnia, psychosis) and gastrointestinal AEs [96]. A retrospective analysis of 1704 patients found 12/24 month neuropsychiatric AE rates of 5.6/6.7% for DTG, 1.9/2.3% for RAL, and 0.7/1.5% for elvitegravir (EVG). For DTG, the most common neuropsychiatric AEs were insomnia, sleep disturbances, dizziness, and painful paresthesia. The rates of neuropsychiatric AEs that led to discontinuation of DTG therapy were higher in female subjects (10%) and patients older than 60 years of age (11%). The hazard ratio for neuropsychiatric AEs leading to discontinuation among female patients (versus male gender) was 2.64 (95% confidence interval: 1.23–5.65, *p* = 0.01) and among older patients (>60 years versus younger age) was 2.86 (95% confidence interval: 1.42–5.77, *p* = 0.003) [97]. However, an analysis [98] of data across five Phase III clinical trials comprising 3353 patients (including three double-blinded, active-controlled studies) found similar neuropsychiatric AE rates (insomnia, anxiety, depression, and suicidality) for DTG, RAL, darunavir, ritonavir-boosted darunavir, ATV, and even EFV, except that insomnia was higher with DTG than with EFV in the SINGLE trial. Overall, insomnia was the most common neuropsychiatric comorbidity. Moreover, neuropsychiatric-related withdrawals were lower with DTG (0–0.6%) than with RAL (0–2.5%). Most cases of suicidality (self-injury, suicide ideation, suicide attempt, suicide completion) among the DTG treatment groups were not considered to be drug-related, such as in patients with a psychiatric history at baseline that increased their risk for suicide. Consistent with these results, an analysis of spontaneously reported AE data from the cohort for HIV patients in the ViiV Healthcare “Observational Pharmaco-Epidemiology Research & Analysis” (OPERA) reported a similarly low prevalence of suicidality and neuropsychiatric AEs (anxiety, depression, insomnia) among patients on DTG, EFV, RAL, or darunavir. Of these groups, anxiety and depression were highest among RAL patients, and insomnia was lowest among darunavir patients. One EFV patient was reported to discontinue treatment as a result of suicidality [98].

Overall, the safety of ETR, RPV, RAL, and DTG have compared favorably to other antiretroviral agents. Categories of better safety include lipid metabolism, glycemia, and bone health. Low liver toxicity that is similar to placebo has been seen for ETR, and RPV liver toxicity is low and comparable to that of EFV [99]. Some clinical results are summarized below.

### 4.2. ETR and RAL

ETR has been associated with a safer profile than EFV and other antiretroviral therapeutic agents. In the double-blinded “SENSE” trial, 157 treatment-naïve patients were randomized between ETR and EFV, while on a background NRTI regimen (either abacavir/lamivudine, zidovudine/lamivudine, or tenofovir/emtricitabine). At week 48, compared to the ETR group, the EFV patients saw significantly larger elevations in total cholesterol (+0.61 mM/L, *p* < 0.0001), triglycerides (+0.33 mM/L, *p* = 0.03), and low-density lipoprotein (LDL)-cholesterol (+0.35 mM/L, *p* = 0.005), although high-density lipoprotein (HDL)-cholesterol was also higher (+0.15 mM/L, *p* = 0.004). The EFV group saw more Grade 3–4 increases in total cholesterol, LDL-cholesterol, and triglycerides (eight patients, six patients, and two patients, respectively) than the ETR group (two patients, one patient, and no patients, respectively) [100]. In a 48-week study in which 43 virologically suppressed patients on a triple therapy that included a PI were randomized between the PI and ETR, significant improvements were only observed in the ETR group when compared to baseline, including declines in cholesterol (from 207 to 191 mg/dL, *p* < 0.001), in serum triglycerides (from 186 to 132 mg/dL, *p* < 0.001), and in glycemia (from 97 to 93 mg/dL, *p* = 0.03) [48].

An analysis of the FDA Adverse Event Reporting System found a higher incidence of myopathy and rhabdomyolysis events for RAL than EFV, ETR, nevirapine, ATV, or darunavir [93]. However, the safety of ETR/RAL dual therapy has been favored by small observational studies. A prospective cohort study of 25 virologically suppressed patients who switched from various cARTs to ETR/RAL dual therapy saw significant improvements from baseline at week 72, including 27.78% declines in triglycerides (*p* = 0.02), reductions of 15.61% in total cholesterol/HDL ratio (*p* = 0.001), a 6.96% increase in HDL cholesterol (*p* = 0.02), and a 7.06% drop in glycemia (*p* = 0.05), as well as improvements in the glomerular filtration rate. All of the patients switching off of EFV regimens saw neuropsychiatric improvements [74]. In another prospective study, 25 virologically suppressed cARTs patients who switched to ETR/RAL dual therapy saw significant drops from baseline in total cholesterol (−17 mg/dL, *p* = 0.01) and triglycerides (−42 mg/dL, *p* = 0.01) at week 48 [77]. An observational study with 38 virologically suppressed cARTs patients switching to ETR/RAL reported significant declines at week 48 in the total cholesterol (−44.3 mg/dL), triglycerides (−81.2 mg/dL), and prevalence of tubular proteinuria (−30.2%). Bone health also improved, with significant increases in phosphoremia (0.52 mg/dL) as well as lumbar (+6.5%) and femoral neck bone mineral density (4.7%) [73].

### 4.3. RPV and DTG

A number of investigations observed a safer lipid metabolism profile with RPV, compared to other NNRTIs. A retrospective, multicenter, cohort study (“ANRS CO3 Aquitaine Cohort 2012–2014”) reported that 304 patients switching from various cARTs to a RPV/emtricitabine/tenofovir disoproxil fumarate regimen saw significant six-month improvements (*p* < 0.001) in fasting total cholesterol (−19 mg/dL), LDL cholesterol (−12 mg/dL), and triglycerides (−27 mg/dL) when compared to baseline, but little change in the total cholesterol/HDL ratio [95]. Other studies have associated improved lipid metabolism with RPV, compared to EFV or to baseline after switching from other therapies [101,102,103].

There has been some debate regarding whether DTG and RPV therapies may be associated with partial impairment of renal function. The above-cited ANRS CO3 Aquitaine Cohort 2012–2014 study observed a significant (*p* < 0.0001) reduction from baseline in the estimated glomerular filtration rate (eGFR) of −11 mL/min./1.73 m^2^. This was not a controlled study [95]. The active-controlled, double-blinded, randomized, Phase III “ECHO” trial, which compared RPV and EFV treatment, observed a small decrease from baseline in eGFR in RPV patients that remained within normal limits [52]. The Phase III, parallel-group, open-label SWORD-1, and SWORD-2 trials, which were also active-controlled, observed no change in eGFR in either a RPV/DTG group or a cARTs control group [78]. It has been shown that DTG is a benign inhibitor of organic cation transporter 2 [104], which helps to facilitate the clearance of serum creatinine [105]. An open-label, placebo-controlled, randomized study with 34 healthy adults observed a 10–14% reduction in creatinine clearance in the DTG group without changing either eGFR or effective renal plasma flow. [104]

The current data support the view that bone mineral density, lipid metabolism, and glycemia are healthier with RPV/DTG than conventional cARTs. The SWORD-1 and SWORD-2 trials observed improved bone health in patients who switched from conventional cARTs to RPV/DTG. When compared to cART patients, the mineral density of RPV/DTG patients increased significantly more in total hip bone (*p* = 0.014) and lumbar spine bone (*p* = 0.039), while the bone remodeling biomarkers decreased, including bone-specific alkaline phosphatase (*p* < 0.001), procollagen type 1 *N*-propeptide (*p* < 0.001), and type 1 collagen cross-linked C-telopeptide (*p* < 0.019) [106]. An observational study following 145 patients for 101 weeks who switched from conventional cARTs to RPV/DTG reported significant reductions in LDL/HDL ratio (*p* = 0.04), as well as significant improvements in total cholesterol, LDL, and glycemia [80].

## 5. Adherence

Non-adherence to treatment remains a problem for HIV-infected patients [107,108], but high potencies and low dose with ETR, as well as low dose/low pill burden with RPV and INSTIs like DTG could lead to improved adherence. An 80-patient prospective study in 2012 reported that changing to an ETR-containing regimen from other cARTs significantly reduced regimen complexity (*p* = 0.035) and patients’ perception of complexity (*p* = 0.015), leading to a significant increase in adherence from 65% to 81.3% (*p* = 0.002) [109]. Switching from 2–3 pills/day to one pill/day has been shown to significantly improve adherence [110,111,112]. Herein lies a key advantage of RPV over ETR, and of DTG over RAL. The adult dose for RPV is 25 mg once/daily [52], compared with the ETR dose of 200 mg twice/daily [57,58,100]. DTG, dosed at 50 mg once/daily imposes a lower pill burden than 400 mg twice/daily RAL [83].

A study [113] with 2541 patients comparing RPV, ETR, EFV, RAL, lopinavir, ATV, and darunavir found that RPV had the lowest discontinuation rate, while RAL was associated with early discontinuation. After three treatment years, the discontinuation rate among treatment-experienced patients was 22% for RPV, 47% for ETR, and 41% for RAL. Using darunavir as a reference, the adjusted hazard ratio for discontinuation among treatment-experienced patients was 0.66 for RPV (confidence interval: 0.52–0.83, *p* < 0.001), 1.61 for ETR (confidence interval: 1.31–1.98, *p* < 0.001), and 1.35 for RAL (confidence interval: 1.15–1.58, *p* < 0.001). Using EFV as a reference, the adjusted hazard ratio for discontinuation among treatment-naïve patients was 0.33 for RPV (confidence interval: 0.20–0.54, *p* < 0.001) and 1.47 for RAL (confidence interval: 1.12–1.92, *p* = 0.005). Thus, patients on RPV-containing treatments discontinued therapy less often than those on ETR or EFV [113], which is in agreement with the findings of the ECHO/THRIVE trials [47,52].

Switching from conventional triple-therapies to a safe and effective dual therapy might further improve treatment adherence. An observational study following 30 patients who switched from conventional cARTs to various dual therapies reported that adherence increased from 86.79% before switching to 96.27% [114], but statistical significance was not reported. In an observational study with 35 patients on various cARTs for a median of 14 years, the proportion of subjects with >90% adherence increased significantly from 65.6% to 93.8% (*p* = 0.004) upon switching to RPV/DTG for 48 weeks [79].

Non-adherence to treatment leads to a rebound of viremia and the emergence of viral drug-resistance mutations. A 2005 study in Italy with 543 subjects found a significant (*p* < 0.0001) linear correlation between reported adherence and the six-month virologic failure rate, defined as a viral load >500 HIV RNA copies/mL. Using this definition, the virologic failure rate was 2.4% for >95% adherence, 4.3% for 86–95% adherence, 12.2% for 76–85% adherence, and 17.4% for ≤75% adherence [41]. An analysis of five studies with 2027 patients from Malawi, Uganda, Kenya, and Cambodia showed a significantly increased risk of drug resistance as adherence declined. When compared to ≥95% adherence, the calculated odds ratio for resistance was 1.79 (95% confidence interval: 1.16–2.76) for 81–95% adherence and 2.91 (95% confidence interval: 1.60–5.29) for ≤80% adherence (*p* = 0.0011 for both). Put differently, poorly or moderately adherent patients were approximately 2–3 times more likely to develop drug resistance than patients with excellent (≥95%) adherence. Patients in these studies presented HIV with resistance mutations to NRTIs and NNRTIs, including ETR [40]. The findings of other studies also support the connection between non-adherence and the emergence of HIV drug resistance [40].

## 6. Drug Resistance

ETR, RPV, and DTG remain active against HIV variants with a variety of mutations that confer resistance to previous generation anti-AIDS drugs. However, new mutations have emerged in treatment-experienced patients that confer resistance to these inhibitors.

### 6.1. Resistance to DAPYs

DAPY compounds ETR and RPV maintain nanomolar or subnanomolar inhibitory activity against HIV-1 RT variants with solitary or pairs of mutations that confer substantial resistance to the first- and second-generation NNRTI drugs delavirdine, nevirapine, and EFV (Table 1) [16,29,44]. Therefore, ETR and RPV were once described to have relatively high genetic barriers to emergence of drug-resistance mutations, leading to long in vitro breakthrough times of resistant viral strains under selection pressure from the NNRTI [59,115,116]. However, HIV variants infecting treatment-experienced patients may already harbor common NNRTI-resistant mutations, which lowers the effective genetic barrier to acquisition of resistance to later-generation NNRTIs, such as ETR or RPV [29]. RPV has been reported to have a lower genetic barrier to emergent drug resistance than ETR [88].

Resistance to RPV might develop more rapidly than EFV. A Phase IIIb, randomized, open-label study (“STaR”, *n* = 786) with 27 subjects selected for the resistance analysis population observed emergent RPV resistance that was more frequent than EFV resistance at week 48. The head-to-head comparison between an RPV treatment group and an EFV treatment group, both with a background treatment of emtricitabine/tenofovir disoproxil fumarate, saw resistance mutation rates that depended on the baseline viral load. For baseline viremia ≤ 100,000 HIV RNA copies/mL, 1.9% of RPV patients developed resistant isolates, compared to 0.8% of EFV patients (*p* = 0.45). For viremia >100,000 copies/mL, the resistance frequency was 9.0% in the RPV group and 0.7% in the EFV group (*p* = 0.001). The comparison between the two groups should be interpreted with caution due to the small sample sizes (20 RPV patients versus seven EFV patients) [117]. Among subjects in virologic failure at baseline (viremia > 100,000 copies/mL) in the Phase III ECHO/THRIVE trials, 17% of 318 RPV patients were in virologic failure at week 48, compared to 7% of 352 EFV patients. Phenotypic resistance data were available for a subset of these populations. Of these, the proportion with phenotypic resistance to RPV (63% of 46 subjects) was larger than the proportion of EFV subjects with phenotypic resistance to EFV (38% of 16). By comparison, patients who were virologically suppressed at baseline (≤100,000 copies/mL) may have shown resistance to RPV less frequently (13% of 16) than to EFV (50% of 12 subjects) [61].

Cross-resistance between NNRTIs has been reported in various studies, and NNRTI-treatment-experienced patients are more likely than treatment-naïve patients to carry HIV variants resistant to ETR and RPV. Among patients that were previously treated with NVP or EFV, mutations conferring resistance to RPV may be more common than mutations conferring resistance to ETR [118]. In a Thailand study of patients in virologic failure following treatment with NVP- or EFV-containing therapies, 60% of 225 adults carried high-level ETR resistance, including some patients with RPV resistance that was highly cross-resistant with ETR [119]. The Phase III ECHO/THRIVE trials saw a high incidence of cross-resistance (93%) to ETR in the virologic failure RPV group [61]. Other studies have reported cross-resistance between ETR and RPV [59,117,120]. The emergence of resistance to DAPYs notwithstanding, systematic reviews of literature covering 52,680–64,466 patients worldwide reported in 2016 that incidence of HIV-1 RT drug resistance to RPV was low (<0.1%), except for E138A/G/K/Q/R (0.7–6.1%) and Y181C/I/V (0.3%) [121].

Mutations that result in resistance to ETR and RPV have been observed in HIV isolates from both adults [40,74,117,120,124,125] as well as children and adolescents [64,126], where higher non-adherence may lead to a higher frequency of virologic failure. Drug-resistant strains that have been seen in children have often shown partial susceptibility to ETR and RPV [126]. Many single mutations are susceptible to ETR and RPV, but they confer drug resistance in certain combinations [74,117,127]. For example, HIV-1 RT with a single V179F mutation is highly susceptible to ETR, showing an FC (fold change from wild-type inhibitory activity) <0.2, and Y181C remains partially susceptible (FC = 4.0–5.1), but Y181C + V179F is highly resistant (FC = 158.9), and Y181C + N348I + T369I is substantially resistant (FC = 51.2) [59,123]. Similarly, susceptibility to RPV has been observed in HIV-1 RT with a single K103N mutation [59,127,128], but the combination of K103N and Y181I is highly resistant to RPV (Table 2) [59]. The resistance profiles of ETR and RPV are not identical. K101P results in relatively low-level resistance to ETR (FC = 5.3) and substantially greater resistance to RPV (FC = 51.7). For comparison, K101P-associated resistance is still higher against EFV (FC = 72.3) and nevirapine (FC > 166.1). Combining K101P with K103N and V108I confers high resistance to RPV (FC > 162.1), but moderate resistance to ETR (FC = 18.4) [59]. Combinations of NNRTI-resistance mutations have been common among RPV patients in virologic failure [117,126]. HIV-1 RT mutations in the NNRTI-binding site that result in resistance against ETR and/or RPV in certain combinations have included V90I, A98G, L100I/V, K101E/P/Q/R/T, K103N/S, V106A/I, V108I, E138A/G/K/Q/R/S, V179F/I/L/D, Y181C/I/V, Y188H/L, G190A/C/E/Q/S/T/V, H221Y, P225H, F227C/L, and M230I/L (Figure 2) [40,59,61,64,74,117,118,120,121,124,125,126,127,129,130,131,132,133]. V179F + Y181C + F227C is an example of a combination of mutations that confers high resistance to both ETR and RPV (Table 2) [59].

Doravirine (DOR), a once/daily NNRTI that the FDA approved in 2018, reference [135] is reported to be more active than RPV against RT variants harboring mutation combinations that include K103N, but there is disagreement in the literature regarding whether other RPV-resistant mutation combinations are more susceptible to DOR [66,136,137]. It has been suggested that RPV and DOR might be compatible as part of a combination therapy due to different efficacy profiles against drug-resistant mutant viruses [137].

### 6.2. Resistance to INSTIs

Although seen less frequently than NNRTI-resistance mutations, INSTI-binding site mutations that confer resistance have included H51Y, T66A/I/K, V72I, L74I/L/M, E92G/Q, Q95R, T97A, G140A/C/S, E138A/K/T, Y143C/H/R/S, Q146R, Q148H/K/N/R, V151I, S147G, S153F, N155H/S, E157Q/K, G163K/R, D232N, and R263K (Figure 3) [131,138,139,140,141,142,143,144,145]. Current indications are that DTG and bictegravir (BIC) have higher genetic barriers to emergent resistance than CAB, EVG, or RAL [21,85,88,138,146,147]. In vitro drug resistance was found to develop much more slowly with DTG and BIC than CAB and EVG [138], and emergence of drug-resistance mutations under selection pressure from DTG is rare [146]. Typically, resistance has been associated with mutations of Y143, Q148, or N155, each along with secondary mutations [141,142]. There are clinical indications that a rare G118R mutation plus accumulation of other mutations can cause resistance to DTG and other INSTIs, leading to virologic failure. However, an in vitro investigation could only produce low-level resistance to DTG (FC = 6.5–8.0), EVG (FC = 6.6–10.0), and RAL (FC = 4.6–11.3) from G118R + T66A + E138K or G118R + T66A + L74M + E138K + V151I. Limited resistance to DTG (FC < 3) is reported for isolates with the S153Y + R263K mutations [148]. Emergent mutations that lead to low-level DTG resistance include H51Y, S153F/Y, R263K, and M50I [138,146]. E157Q with certain secondary mutations leads to increased susceptibility to DTG, BIC, and CAB, but resistance to EVG and RAL (Table 3) [138].

Some INSTI-resistance mutations associated with virologic failure (E92Q, N155H, G140S, Q148H, and E157Q, R263K) have been shown by in vitro assays to impair IN activity substantially in the absence of inhibitor. The accumulation of additional mutations can compensate for impaired fitness of some viral variants. The G140S + Q148H double mutant can rescue IN activity that is impaired in the solitary G140S and Q148H mutants [142,145,146,152].

In phenotypic assays of HIV-1 isolates from virologically failing patients on RAL-containing therapies, it was shown that DTG retains full or partial inhibitory activity against RAL- and EVG-resistant HIV variants in both CD4^+^ T-cells and macrophages. Mutations resistant to RAL or EVG (FC ranging from 4.3 to > 100) that were highly susceptible to DTG (FC ranging from 1.0 to 2.0) included T97A + Y143C/R, N155H, G140S, and Q148H (Table 3). Some combinations of mutations that included Q148H (e.g., G140S + Q148H, T92A + G140S + Q148H, and G140S + Q148H + G163R) were partially susceptible to DTG and highly resistant to RAL and EVG [141,147]. Variants with the G140S + G147S + Q148K mutation profile were moderately or highly resistant to CAB, EVG, and RAL, but susceptible to DTG and BIC. Q148 mutation combinations leading to low or moderate resistance to DTG and BIC, but high resistance to CAB, EVG, and RAL included L74I + E138K + G140S + Q148R and L74M + E138K + Q148R + R263K. The combination of L74M + G140S + S147G + Q148K was associated with high resistance to all five INSTIs (Table 3) [138]. In general, DTG and BIC are active against HIV variants carrying IN mutations that result in moderate or high resistance to RAL, EVG, and/or CAB [138,141,146,147,149].

## 7. Mechanism of Inhibitor Response to Drug-Resistance Mutations

NNRTIs bind to HIV-1 RT in a hydrophobic pocket near the catalytic site of the enzyme’s DNA polymerase activity. This flexible binding site is transiently created as the enzyme samples its various conformations [153,154]. Most NNRTI-resistance mutations directly antagonize binding of various NNRTIs in this pocket (or increase the dissociation rate with respect to association rate), while secondary mutations improve the fitness of drug-resistant variants of RT (Figure 2) [16,29,155,156,157]. Direct antagonistic mechanisms include steric hindrance, the removal of important protein-inhibitor interactions, and blocking entry of the NNRTI to the binding site [158]. The key to the resilience of the DAPY compounds in the face of common NNRTI-resistance mutations (Table 1) resides in the “strategic flexibility” of their chemical structures (Figure 4), which consist of three aromatic moieties that are linked by pairs of single bonds with rotational freedom (Figure 1a). Rilpivirine adds a cyanovinyl substituent that is linked to one of the aromatic rings by an additional single bond. X-ray crystal structures of ETR and RPV in complex with wild-type and mutant RTs have shown that small to moderate torsional rotations about these single bonds allow the compounds to adjust their conformations and positions either to maintain important interactions with RT variants or to replace lost contacts with compensatory ones, whenever mutations are introduced that challenge the compounds with steric hindrance or change the electrostatic environment of available interactions (Figure 5) [29,42,44,159]. X-ray crystal structures suggest that DAPY analogs use similar strategic flexibility to maintain inhibitory activity in response to mutations at the NNRTI active site and provide strong support for predictions regarding the response of DAPYs to mutations at the NNRTI binding pocket [160,161]. The DAPY-class NNRTIs show varying levels of vulnerability to K101 and K103 mutations, which are believed to hinder entry to the NNRTI binding pocket [158] via mechanisms that may include (1) changes in the geometry of this putative pocket entrance by abolishing a salt bridge between K101 and E138 [42] and (2) changes in hydrogen-bonding networks [162].

In typical INSTIs, a triad of coplanar oxygen and/or nitrogen atoms provide lone pair electrons that chelate two divalent cations (Mg^2+^ or Mn^2+^) at the catalytic site of IN, while simultaneously binding a halogenated aromatic ring in a pocket that is natively occupied by a 3’-terminal deoxyadenosine of the reverse-transcribed viral DNA [67,69,70]. In DTG (Figure 1b), these components comprise three oxygen atoms on a tricyclic pyridinone scaffold and a difluorobenzyl ring, bridged by a flexible linker with two freely rotatable single bonds, as well as two more single bonds in an enolate π-conjugated to the planar aromatic portion of the pyridinone ring system [67,71]. Although the π-conjugation limits the rotational freedom of two of the four single bonds in the linker, the relative resilience of DTG in the face of INSTI-resistance mutations has been attributed, in part, to the overall flexibility of this linker as well as its length. X-ray crystal structures of DTG bound to wild-type and drug-resistance mutation-bearing active sites have shown that DTG can adjust its position and conformation to remain bound when certain resistance mutations are introduced (Figure 6) [71]. RAL, CAB, and BIC share the same linker with DTG (Figure 1b), but differ in the oxygen atoms that chelate the divalent metal atoms at the IN active site. In these other INSTIs, the flexible linker’s amide oxygen atom participates in the chelation of the metal atoms. In DTG, the coordination of the metal cations might be more optimal, because all of the chelating oxygen atoms are part of the rigid coplanar aromatic portion of the tricyclic ring system. In addition, freeing up the linker amide oxygen allows for DTG to sit deeper in the binding pocket, leading to stronger protein interactions with the halogenated benzyl ring [67,71]. These structural conclusions should be viewed with caution, however, because structures of INSTIs bound to IN are not yet available, and structural observations are based on IN-orthologous prototype foamy virus (PFV) intasomes in complex with inhibitors.

## 8. Exciting Prospects for Long-Acting Treatments

Among the greatest obstacles to controlling HIV/AIDS is the need for patient compliance with medical regimen, a complicated problem even with the advent of highly effective treatments. New long-acting treatments are under development in order to improve regimen adherence and minimize the emergence of drug resistance (for review, see refs [163,164]). These treatments are based on formulations that are administered either orally or by injection weekly, monthly, or at eight-week intervals. A proof-of-concept investigation showed that a novel orally-administered capsule design could release DTG, CAB, and RPV in an in vivo pig model so that each drug had a sustained plasma concentration of ~200–800 ng/mL over the course of a week [165]. A randomized, Phase IIb, 286-patient, open-label study (LATTE-2) tested formulations of RPV/CAB that were injected either monthly or every eight weeks [166]. The study used an active control group on oral CAB/abacavir/lamivudine. At week 96, 84% of the control oral group, 87% of the four-week group, and 94% of the eight-week group were virologically suppressed. Serious AEs that were unrelated to the administered drugs were reported in 10% of patients receiving the intramuscular injections and 13% of control group patients. These studies suggest the feasibility of long-acting dual and triple-drug therapies that combine RPV with INSTIs to optimize adherence and avoid virologic failure.

## 9. Conclusions

ETR and RPV have had a major impact in the fight against AIDS. They have been effective in patients harboring HIV variants that are resistant to earlier generation NNRTIs and have been safer to use. Their high potencies have led to substantially lower dosages or pill burdens and improved adherence, which potentially slow the rate of emergence of new resistant viral variants. They have a higher genetic barrier to emergent resistance than some earlier generation NNRTIs, often requiring combinations of mutations for resistance breakthrough to occur.

On the other hand, resistant combinations of mutations have emerged, and RPV patients may develop resistance more often than EFV subjects. NNRTI-resistant viral variants infecting treatment-experienced patients are more likely to develop resistance to ETR and RPV than those of treatment-naïve patients. Patients that switched to RPV-containing therapies who have a high viral load or low CD4^+^ T-cell count show a significantly higher propensity for virologic failure than EFV patients. Although virologic breakthrough appears to be more common with RPV than ETR, RPV has a better safety profile and a lower pill burden. These DAPY-class NNRTIs may be favored for maintenance therapy in patients who have already achieved viral suppression with a previous therapy.

Treatments that combine RPV or ETR with INSTIs such as RAL, CAB, or DTG hold promise. Resistance mutations are still rare with INSTIs. DTG shows a relatively high genetic barrier to resistance. INSTI-resistance mutations tend to be more susceptible to DTG than RAL, CAB, or EVG (Table 3). These INSTIs have relatively good safety profiles. Therefore, combining them with DAPY-class NNRTIs could result in effective treatment coupled with lower AE frequencies than other cARTs. Strategies that may further improve adherence and hinder emergence of resistant strains include dual therapies and long-acting treatments. ETR/RAL and RPV/CAB dual therapies have been studied, and RPV/DTG has been approved. These therapies are non-inferior to conventional treatments. RPV/DTG imposes a lower pill burden than other NNRTI/INSTI dual therapies tested and may have better prospects for long-term adherence. Exciting work on long-acting treatments (either as dual or triple therapies) has begun and suggests that this approach might be feasible in the future. Overall, once patients achieve viral suppression, ETR or RPV in combination with INSTIs may be an attractive option as part of long-term maintenance therapies that optimize efficacy, adherence, and safety.

## Figures and Tables

**Figure 1 pharmaceuticals-13-00122-f001:**
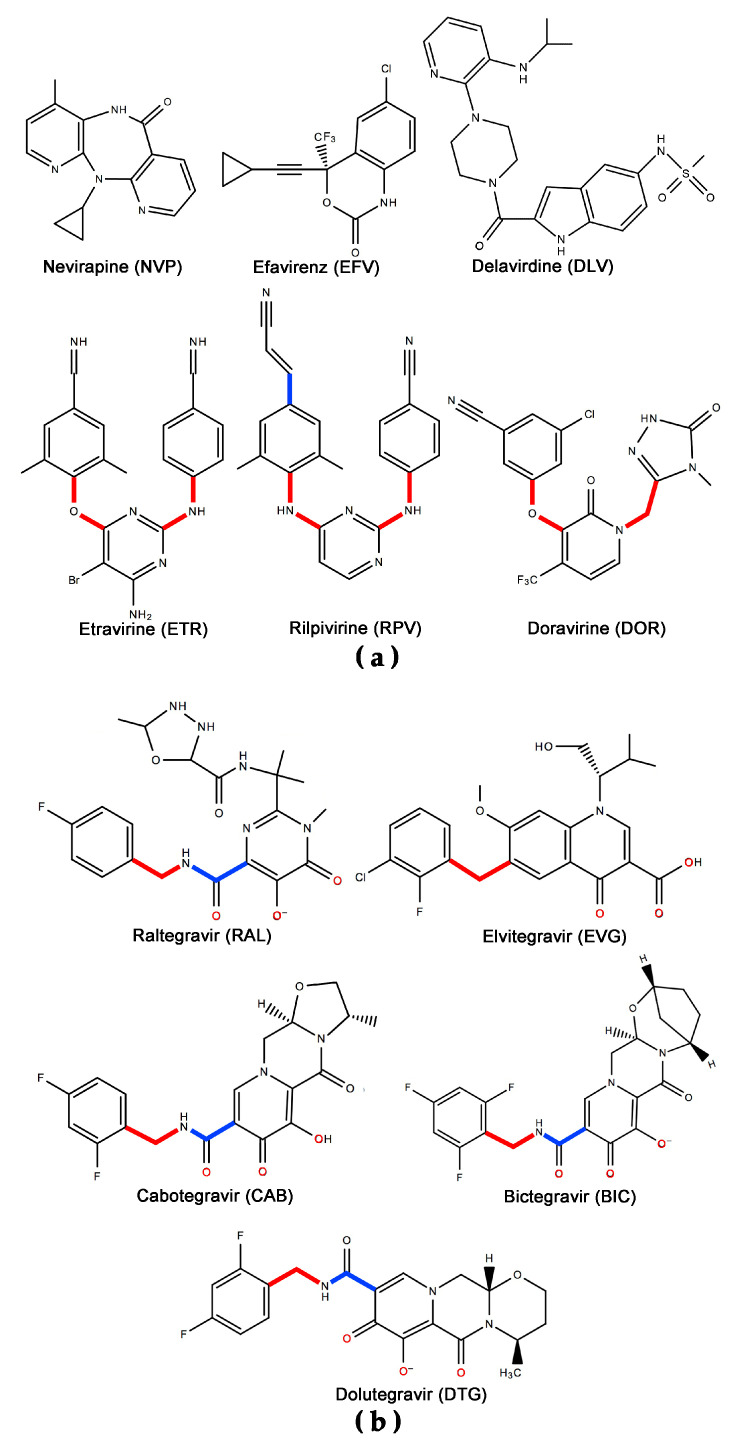
NNRTI-class and IN strand transfer inhibitor (INSTI)-class drugs. Chemical structures are shown for some NNRTIs (**a**) and INSTIs (**b**) currently approved or in trials [42,66,67,68]. Important single bonds with broad rotational flexibility are shown in red or with partial flexibility in blue. Strategic flexibility of some of these compounds plays in an important role in retaining inhibitory activity against HIV enzymes bearing drug-resistance mutations. INSTI oxygen atoms (**b**) believed or shown in crystal structures to chelate pairs of divalent cationic metal atoms at the catalytic site are depicted in red [69,70,71,72].

**Figure 2 pharmaceuticals-13-00122-f002:**
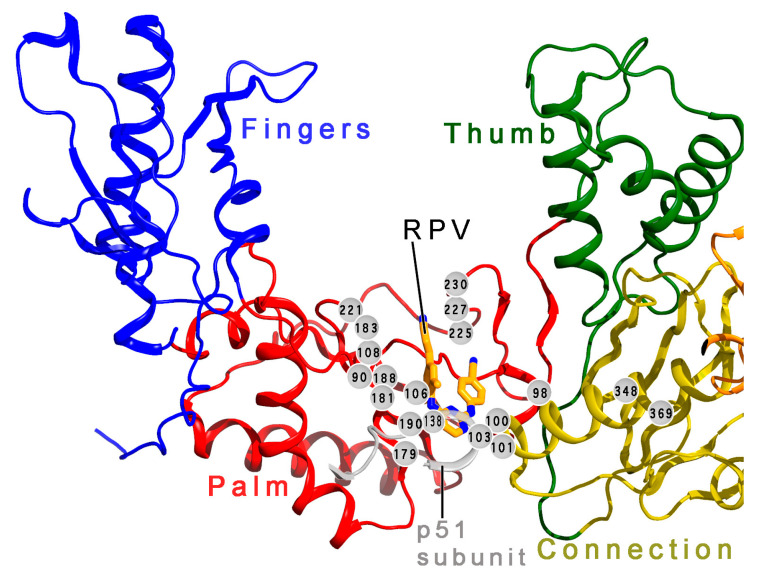
Locations of resistance mutations in RT. The locations of commonly seen NNRTI-resistance mutations (gray orbs with residue numbers) are shown in a ribbon diagram of part of the catalytic p66 subunit of RT and a portion of the p51 subunit. RPV (tan carbon atoms, blue nitrogen atoms) is shown in the NNRTI-binding pocket. Although most resistance mutations occur at or near the binding pocket in the p66 palm subdomain (including at position 138 of the p51 subunit), some mutations are seen in other subdomains of RT and may improve the fitness of drug-resistant variants of RT. The coordinates are taken from the Protein Data Bank (PDB), accession ID 4G1Q [134]. The figure was generated using PyMol^®^, with annotation in Photoshop^®^.

**Figure 3 pharmaceuticals-13-00122-f003:**
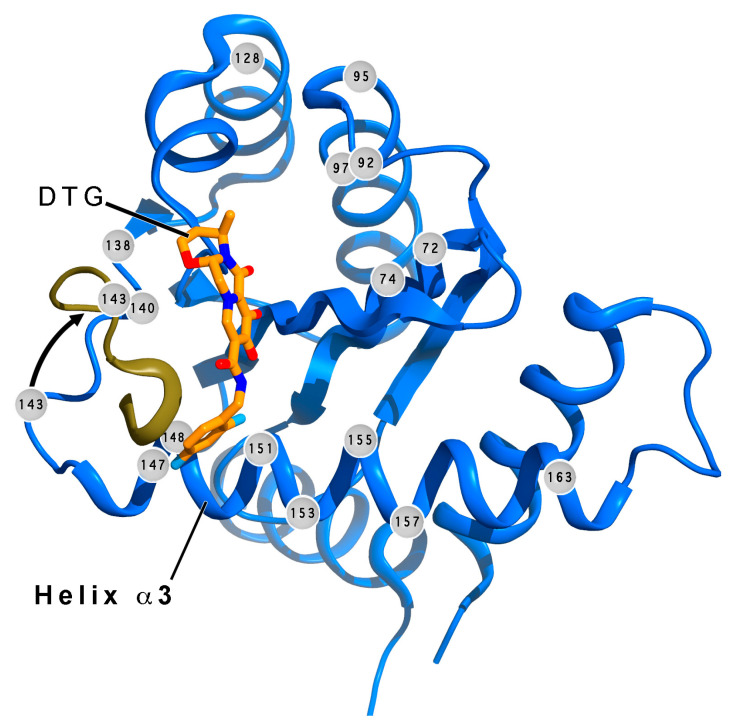
Locations of resistance mutations in IN. The locations of commonly seen IN-resistance mutations (gray orbs with residue numbers) [131,138,139,140,141,142,143,144,145,147,149,150] are shown in a ribbon diagram of the catalytic core of IN (blue, apo-enzyme structure). The shown apo-structure (light blue, PDB accession ID: 4CJE) was superimposed on that of the DTG-bound IN-orthologous prototype foamy virus (PFV) intasome (PDB accession ID: 3S3M). DTG (tan carbon atoms, red oxygen atoms, blue nitrogen atoms, and cyan fluorine atoms) is shown in its binding pocket. The loop (gold) containing the Y143C/H/R mutation site (magenta) rotates toward the bound DTG (black arrow) in the PFV structure. Many mutation sites are seen on helix α3 (nomenclature from ref. [151]) and at (or near) the INSTI binding pocket. The two structures were superimposed in Coot^®^, and the figure was generated using PyMol^®^, with annotation in Photoshop^®^.

**Figure 4 pharmaceuticals-13-00122-f004:**
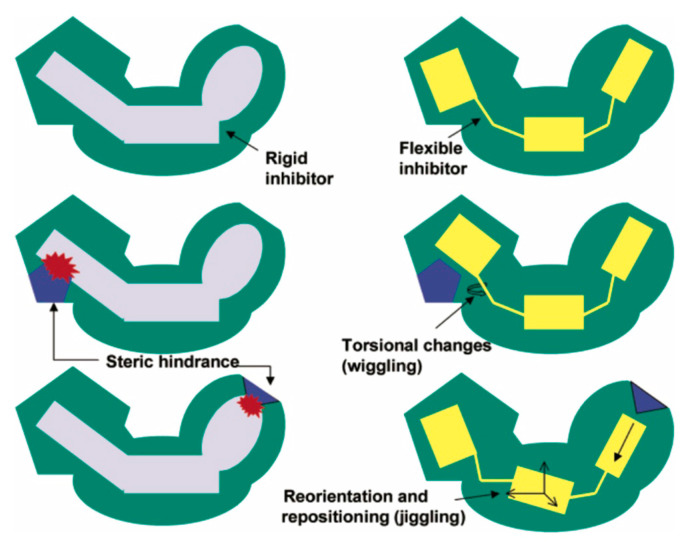
Strategic flexibility. This schematic diagram depicts how a flexible inhibitor can adjust its conformation and position, making it more effective than a rigid inhibitor at overcoming steric hindrance from a resistance mutation at the inhibitor-binding site. Reprinted with permission from Das et al. *J. Med. Chem. 47:2550–2560* ©2004 American Chemical Society·[44].

**Figure 5 pharmaceuticals-13-00122-f005:**
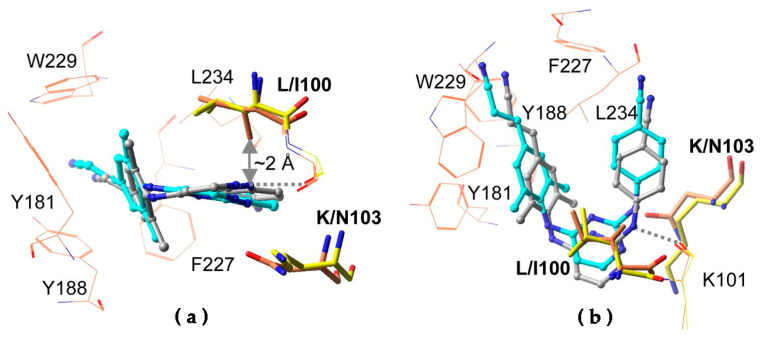
Strategic flexibility in RPV. Differences in the conformation (**a**) and position (**b**) of RPV in the NNRTI binding pocket of wild-type RT (gray) and L100I/K103N RT (cyan) are shown. Reprinted with permission from Das et al. Proceedings of the National Academy of Sciences USA. 2008;105(5):1466-1471 ©2008 National Academy of Sciences, U.S.A [42].

**Figure 6 pharmaceuticals-13-00122-f006:**
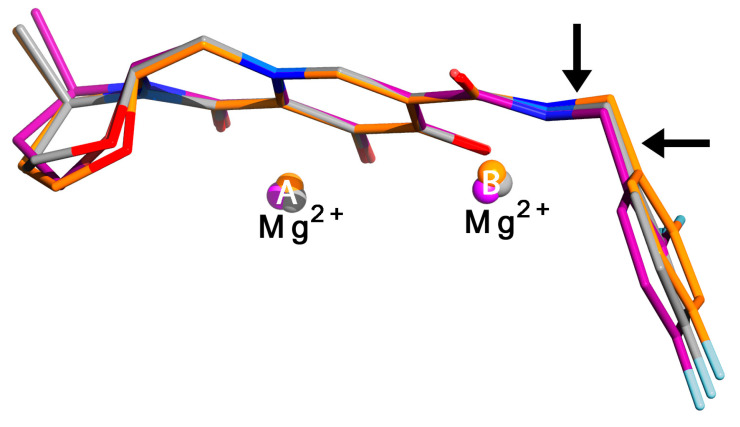
Strategic flexibility in DTG. DTG coordinates are superimposed from three structures of IN-orthologous prototype foamy virus (PFV) intasomes, showing the relative conformations of DTG observed in wild-type PFV (gray carbon atoms) and with resistance mutations corresponding to N155H (magenta carbon atoms, N224H in PFV) or Q148H (orange carbon atoms, S217H in PFV). Black arrows indicate the torsional angles in the flexible linker that are observed to rotate the most. Flexibility is also seen in the non-aromatic portion of the tricyclic ring system (left side of figure). The aromatic rings of the DTG coordinates were manually superimposed in Coot^®^, and the figure was generated using PyMol^®^, with annotation in Photoshop^®^. Coordinates were taken from PDB accession IDs 3S3M (wild-type), 3S3O (PFV N224H), and 3S3N (PFV S217H).

**Table 1 pharmaceuticals-13-00122-t001:** Non-nucleoside reverse transcriptase inhibitor (NNRTI) antiretroviral potency against wild-type and some mutant reverse transcriptases (RTs).

Resistance Mutation	Compounds with Antiretroviral Activity (EC_50_, nM) ^§^				
	Rilpivirine (RPV, TMC278)	Etravirine (ETR, TMC125)	Efavirenz (EFV)	Nevirapine (NVP)	Delavirdine (DLV)
Wild-type	0.4	2	1.4	81–85	16
L100I	0.4–0.5	3	35–38	597–638	3467
K103N	0.3	1	28–39	2467–2879	1697
Y181C	0.1–1.3	6	2	5351–10,000	1336
K103 + Y181C	1.0 ^§^–1.4 *	5.0 ^§^–8.2 *	37.0 ^§^–51.1 *	≥10,000	>10,000
Y188L	2.0	3.0	78–138	≥10,000	178
Legend	<1.0 Highly active	1.0–10 Active	10.1–100	High Resistance (>100)	

^§^ Antiretroviral activities taken from various reported in vitro cell assays [29,42,43,44]. * Activity estimated from “Table 3” in Azijn et al. [59] by multiplying the given fold change by the wild-type activity in this table. Color coding: Light blue, highly active (EC_50_ < 1.0); yellow, active or low-level resistance (1.0 ≤ EC_50_ ≤ 10); tan, moderate resistance (10.1 ≤ EC_50_ ≤ 100); red, high resistance (EC_50_ > 100).

**Table 2 pharmaceuticals-13-00122-t002:** NNRTI resistance from selected emergent HIV-1 RT mutations.

	Fold Change ^§^			
NNRTI-Resistance Mutations	Rilpivirine	Etravirine	Efavirenz	Nevirapine
Y181C	2.7	4.0–5.1	2.1	>43.0
Y181C + V179F	8.7	158.9	4.6	>358.3
Y181C + N348I + T369I		51.2 ^‡^	17.0	>400 ^‡^
K101P	51.7	4.36 ^‡^–5.3 ^§^	72.3	>166.1
K103N	0.9	0.9 ^§^–1.28 ^‡^	21.3 ^‡^–32.5 ^§^	>42.1
K103N + Y181I	94.9	16.1	6.4	>71.5
L100I + K103N + V179L	46.1	13.4	5,660.60	>71.5
L100I + K103N + Y181C	80.8	58.1	1,812.00	468.1
K101P + K103N + V108I	>162.1	18.4	12,931.10	>51.6
V179 + Y181C + F227C	553.8	638.6	25.7	>71.5
Legend	Low-Level Resistance (1.0 = Wild-Type Activity)	Resistance (10–50)	High Resistance (>50)	Data Unavailable

^§^ Selected mutants conferring resistance to RPV and/or ETR are shown. Fold change (FC) was calculated as the (EC_50_ of site-directed mutant RT)/(Wild-type RT EC_50_). 1.0 = wild-type activity. Median FC is given except as indicated. See Azijn et al. and Gupta et al. for a more complete list of NNRTI-resistance mutations data [59,122,123]. All mutations tested have been reported clinically. ^‡^ Mean FC is reported. Color coding: White, data unavailable; yellow, susceptible or low-level resistance (1.0 ≤ FC < 10); tan, moderate resistance (10 ≤ FC ≤ 50); red, high resistance (FC > 50).

**Table 3 pharmaceuticals-13-00122-t003:** INSTI resistance from emergent HIV-1 integrase mutations.

Emergent INSTI-Resistant Mutations	Fold Change ^§^				
	Dolutegravir (DTG)	Bictegravir (BIC)	Cabotegravir (CAB)	Elvitegravir (EVG)	Raltegravir (RAL)
T66I + Q95K + E157Q	0.8	0.6	0.9	89	4.7
T66I + Q95K + E157Q + S230R	0.1	0.01	0.1	156	3.2
T97A + A128T + E157Q + V151I	0.6	0.3	0.5	52	21
T97A + Y143R	1.0–1.2			1.9–2.1	11.6–60.4
T97A + Y143C	1.0–1.5			4.3–4.8	44.7– >100
N155H	1.0–1.8	1.6–1.8	1.0–2.1	7.0–8.5 ^ab^	20–47.8 ^b^
G140S	0.86–1.3		0.81	2.7–9.7	1.1–1.8
Q148H	0.97–1.3		0.86	7.3–9.3	13–16
G140S + Q148H	2.6–11.0	4.8	6.1–16.8	38.9–>100	>100
T92A + G140S + Q148H	5.5–18.3			>100	>100
G140S + Q148H + G163R	10.0–18.4			>100	>100
Q148K	1.1–2.0	2.2	3.1–5.6	11.8 ^c^	3.2 ^c^
G140S + G147S + Q148K	4.7	1.8	40	720	60
E138K + Q148R + R263K	14	8	8.3	>100	7
L74I + E138K + G140S + Q148R	25	5.3	87	57	>100
L74M + E138K + Q148R + R263K	17–24	12–17	>100	>1500	188–355
L74M + G140S + S147G + Q148K	162	120	700	>1000	900
Legend	Most Susceptible (<1.0)	Susceptible or Low-level Resistance (1.0 = Wild-type activity)	Resistance (10–50)	High Resistance (>50)	Data Unavailable

^§^ Fold change (FC) was calculated as the (EC_50_ of site-directed mutant IN)/(wild-type IN EC_50_). 1.0 = wild-type activity. For a more complete list of INSTI-resistance mutations data, see refs [138,141,147,149,150]. All mutations tested have been reported clinically. There were discrepancies in the literature on FC resistance to EVG and RAL: ^a^ Tsiang et al. reported N155H FC of 48 for EVG [147]. ^b^ Yoshinaga et al. reported N155H FC’s of 25 for EVG and 8.4 for RAL [150]. ^c^ Yoshinaga et al. reported Q148K FC’s of >1700 for EVG and 83 for RAL [150]. Color coding: White, data unavailable; light blue, more susceptible to inhibition than wild-type (FC < 1.0); yellow, susceptible or low-level resistance (1.0 ≤ FC < 10); tan, moderate resistance (10 ≤ FC ≤ 50); red, high resistance (FC > 50).

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
