# Peer review of "Non-Nucleoside Reverse Transcriptase Inhibitors Join Forces with Integrase Inhibitors to Combat HIV"

_pharmaceuticals, 2020, doi:10.3390/ph13060122_

Round 1

Reviewer 1 Report

This is well-researched paper with a clear structure. I only feel qualified to comment on the more epidemiological aspects of this paper – the chemistry is well beyond my expertise and I would advise obtaining an expert review in this field before publication.

 I only have a few comments:

  • Table 1 needs a clearer explanation of what it is displaying.
  • All Tables should have a key to explain what the colour coding means.
  • In the Safety section, the authors report (on p5 line 191 and p6 line 207) on hazard risks when they mean hazard ratios.

Reviewer 2 Report

Himmel and Arnold reviewed and comment on the recently published data (including clinical as well as safety, adherence and efficacy data from several CTs) on the use of the newest NNRTIs analogs, known as the diarylpyrimidine (DAPY) analogs etravirine (ETR) and rilpivirine (RPV) combined with or not with the INSTIs and mainly bictegravir (BIC) and dolutegravir (DTG) that perform better against a wide range of resistance HIV variants than other NNRTIs/INSTIs. The review describes, summarises and comments on the usage of these new anti-HIV compounds in terms of safety, adherence and effectivity especially in patients harbouring multidrug resistant HIV variants. Moreover, the review summarises what is known about this anti-viral agents in terms of drug resistant associated mutations and the mechanisms involved acquisition of resistance. The authors conclude that DAPY NNRTIs (i.e ETR or RPV) in combination with INSTIs may be treatments of choice for long-term maintenance therapies with optimized efficacy, adherence, and safety. Overall, the review is nicely presented, well-structured and well written.    

Minor suggestion

  • In the introduction segment (line 39-41), the authors state that “A key goal of treatment is to maintain a low viral load in the plasma.”  Nowdays, (i.e in the U=U era) the key goal is to achieve total viral suppression, i.e undetectable viral load. Moreover, I would suggest the authors to include in the introduction a few infos re the U=U, plus the relevant major studies published in the NEJM, JAMA and Lancet.   
